# A One Health Perspective on the Human–Companion Animal Relationship with Emphasis on Zoonotic Aspects

**DOI:** 10.3390/ijerph17113789

**Published:** 2020-05-27

**Authors:** Paul A.M. Overgaauw, Claudia M. Vinke, Marjan A.E. van Hagen, Len J.A. Lipman

**Affiliations:** 1Department Population Health Sciences, Division of Veterinary Public Health, Institute for Risk Assessment Sciences, Faculty of Veterinary Medicine, Utrecht University, P.O. Box 80178, 3508 TD Utrecht, The Netherlands; p.a.m.overgaauw@uu.nl; 2Unit Animals in Science & Society, Animal Behaviour, Department Population Health Sciences, Faculty of Veterinary Medicine, Utrecht University, P.O. Box 80166, 3508 TD Utrecht, The Netherlands; c.m.vinke@uu.nl (C.M.V.); m.a.e.vanhagen@uu.nl (M.A.E.v.H.)

**Keywords:** One Health, companion animals, pets, human–animal bond, anthropomorphism, zoonoses, hygiene hypothesis

## Abstract

Over time the human–animal bond has been changed. For instance, the role of pets has changed from work animals (protecting houses, catching mice) to animals with a social function, giving companionship. Pets can be important for the physical and mental health of their owners but may also transmit zoonotic infections. The One Health initiative is a worldwide strategy for expanding collaborations in all aspects of health care for humans, animals, and the environment. However, in One Health communications the role of particularly dogs and cats is often underestimated. Objective: Evaluation of positive and negative One Health issues of the human–companion animal relationship with a focus on zoonotic aspects of cats and dogs in industrialized countries. Method: Literature review. Results: Pets undoubtedly have a positive effect on human health, while owners are increasing aware of pet’s health and welfare. The changing attitude of humans with regard to pets and their environment can also lead to negative effects such as changes in feeding practices, extreme breeding, and behavioral problems, and anthropozoonoses. For the human, there may be a higher risk of the transmission of zoonotic infections due to trends such as sleeping with pets, allowing pets to lick the face or wounds, bite accidents, keeping exotic animals, the importation of rescue dogs, and soil contact. Conclusions: One Health issues need frequently re-evaluated as the close human–animal relationship with pet animals can totally differ compared to decennia ago. Because of the changed human–companion animal bond, recommendations regarding responsible pet-ownership, including normal hygienic practices, responsible breeding, feeding, housing, and mental and physical challenges conforming the biology of the animal are required. Education can be performed by vets and physicians as part of the One Health concept.

## 1. One Health in Companion Animals

The One Health initiative or concept is a worldwide strategy that recognizes that public health is connected with animal health and the environment. It concerns multidisciplinary collaboration between physicians, veterinarians, environmental scientists, public health professionals, wildlife experts, and many others [1,2]. With a multisectoral and transdisciplinary approach, public health threats can be better monitored and controlled. The resulting synergism enhances the knowledge of how diseases, known as zoonotic diseases, can be shared between animals and people with the goal of achieving optimal health outcomes [1,3]. One Health is not a new concept, but it has become more important since 2006 as a result of emerging and re-emerging diseases [3].

The concept of One Health is nothing new and it started more than a century earlier with the theme One Medicine by the 19th century German physician and pathologist, Rudolf Virchow. He introduced the term “zoonosis” and did not distinguish a dividing line between human and animal medicine [4]. Other visionary scientists in this field were the Canadian physician and pathologist Sir William Osler, James Steele who developed the discipline of veterinary public health at the Centers for Disease Control (CDC) in the US, and Calvin Schwabe widely known as the father of veterinary epidemiology who wrote the first handbook in 1964: Veterinary Medicine and Human Health. The One Medicine term has evolved into One Health, placing emphasis on health promotion rather than treating diseases [5,6,7].

Many One Health initiatives focus mainly on the relationship between humans and livestock or wildlife health, because several zoonotic disease pandemics and (re)emerging infectious diseases originated from these animal species. Examples of such infections are West-Nile virus, corona virus (SARS, Covid-19), zika virus, avian H5N1 influenza virus, Nipah virus, and Hendra virus [8]. The recently founded One Health European Joint Programme (OHEJP) also focuses on foodborne zoonoses, antimicrobial resistance and emerging threats, while companion animals are absent [9].

However, the role of companion animals, particularly dogs and cats, is often underestimated in One Health communications. During past decades, dogs and cats more often spend their life indoors in very close physical contact with their owners. There are a number of zoonotic infectious diseases, as well as resistant bacteria, that may be transmitted directly or indirectly from these species [5,10]. On the other hand, companion animals may be effective sentinels, as they share a common environment with their owners. They can help in the early identification of food contamination, infectious disease transmission, environmental contamination, and even bioterrorism or chemical terrorism events [11].

The World Small Animal Veterinary Association (WSAVA) One Health Committee considered that there are three key areas of One Health regarding companion animals: (1) The human–companion animal bond, (2) comparative and translational medicine, and (3) zoonotic infectious disease [12].

The relationship between animal health and public health with regard to companion animals, can be described as sharing the same living environment (both indoors and outdoors), being sensitive to a number of similar pathogens—zoonoses, and often treating with the same medicines if infections occur. In the latter, the development of antibiotic resistance plays a role.

The trends of pet ownership will first be briefly explained. Subsequently, the changing positive and negative attitude of humans (pet owners as well as non-pet owners) towards (pet) animals, mainly dogs and cats, and their environment in industrialized countries will be discussed. This change in attitude can lead to a disturbed human–animal relationship resulting in animals with behavior problems and immune suppression due to stress and associated high cortisol blood levels. Such animals are also more sensitive to contract infections and may therefore at higher risk of transmission of pet-associated zoonotic infections to humans. This paper will focus on the zoonotic aspects within the relationship between humans and pets as part of the One Health concept. The problems associated with keeping unusual exotic animals will be briefly discussed. Recommendations to prevent the transmission of zoonotic pathogens from pets will be formulated.

## 2. Methods

This article is based on an existing post-graduate course for veterinarians, vet technicians, family doctors, midwifes, and specialists such as pediatricians which explores healthy human–animal relationships. The existing evidence-based knowledge contained in this course has been actualized by performing a literature search to add new relevant publications. A literature search was conducted through 2 March 2020, using the National Library of Medicine’s PubMed for the terms “One Health” and “companion animals”; “pet ownership”; “households” and “pets”; “dogs” or “cats” or “pets” and “mental” or “physical health” or “children”; “animal assisted therapy”; “dogs” or “cats” and “nutritional problems” or “overweight” or “obesity” or “homemade” or “raw meat diets”; “dogs” or “cats” and “behavior problems” or “aggression” or “fear” or “anxiety” or “abnormal repetitive behavior”; “dogs” or “cats” and “breeding” or “genetic problems”; “dogs” or “cats” and “zooanthroponoses”; “pets” and “anthropomorphism”; “dogs” or “cats” or “exotic animals” or “rescue dogs” or “soil” and zoonoses. For some topics the internet was accessed and used as reference if additional information was not available as scientific publication. The authors selected articles that described pivotal and novel insights in the different topics. All searches were carried out without filters. The titles of all found articles were screened for relevance to the topic, and appropriate titles were assessed and selected based on their abstracts. If a selected article was a review, it was read and relevant citations were used to find primary literature on the subject. Additional studies were found using the bibliographies of selected articles. Occasionally, reviews were directly used as sources, mostly to convey background information that is not in the core focus of this article. Original articles in English and different national languages (Dutch, German, French, Spanish, if available) were included. Specific searches were made for citations dated after the year 2000 to ensure more recent literature on the topic had not been missed.

## 3. Trends in Pet Ownership

A pet or companion animal is an animal that lives in or around the house and is fed and cared for by humans. Until the 1960s, pets were mainly kept as utility animals, for example as draft dogs or watch dogs or for pest control when it comes to cats. Due to major changes that have taken place in society since the Second World War, such as increased leisure time and prosperity, but also individualization of humans, animals are nowadays kept as pets and are regarded by many owners as valued family members, e.g., over 90% in the United Kingdom [13].

Pet ownership is still increasing in many industrialized countries and these animals are more often considered a member of the family [5]. Even in China, a country where pets were banned in urban areas until 1992, pet ownership has grown quite rapidly in the major cities. The rate of pet ownership of all USA households increased from 57.6% to 59% in 2018. Dogs continue to dominate in popularity among American households. Approximately 38% of households nationwide owned a dog, bringing the population of pet dogs to nearly 77 million, while 25% of households owned cats, with a total population of 58 million [14]. In 2018, an estimated 80 million European households owned at least one pet animal; 24% of households owned dogs and 25% owned cats [15]. There are 85 million pet dogs and 104 million pet cats in Europe which is a 15% increase of dogs within 8 years (74 million dogs in 2010) and a 22% increase of cats (85 million in 2010) [16]. Pet cats in Europe thereby are more popular than dogs.

An explanation of the higher popularity of cats may be the number of single-person households in the EU that rose on average by 2.0% per annum between 2006 and 2016 to 32.5%, and the growth of two-family households grew by 1% to 31% in this period [17]. Also, dual-earner families are widespread as a result of quite a steep growth in female employment over the past two decades [18].

## 4. Positive Aspects of the Human–Animal Relationship

### 4.1. Positive Aspects of Humans for Companion Animals

When animals live with humans, they too benefit from human interaction. Over the past decades, animal welfare has evolved to recognize that animals are sentient beings capable of experiencing positive and negative emotions. The social and ethical dimensions of animal welfare, which are concerned with how human society morally regards and treats non-human animals, are also increasingly being recognized [19]. In Dutch law, the intrinsic value of kept animals is expressly incorporated and used as a guiding ethical principle that forms the basis of any further legislation. The intrinsic value is thereby defined that animals are sentient beings that can feel pain and discomfort. Therefore, they should be kept free of stress, pain, disease, hunger, thirst, and should be able to show natural behavior known as The Five Freedoms of R. Brambell 1965 [20].

In industrialized countries animal keepers, their owners, are legally required to provide such circumstances and can be prosecuted if they infringe the law. Of course, in the field there is animal abuse, negative animal welfare conditions, and animal diseases. However, in general, caretaking for companion animals is nowadays performed at a high level. New insights into animal behavior has had its influence on the general public. For example, owners are aware or are told by vets and pet shops that rabbits should not be kept alone but at least in pairs due to their need for social contact [21].

### 4.2. Positive Aspects of Companion Animals for Humans

There are various reasons to keep pets, such as love, warmth, and companionship. Companion animals have an important emotional value, and promote the socialization of the lonely elderly because they facilitate additional contact with people. Pets form a goal in life, reduce stress, and ensure that the owner keeps physically active.

Around 95% of dog owners and 93% of cat owners expressed that owning their pet makes them happy and 44% of owners selected this as one of the reasons they got their pet in the first place [22]. However, the function of companion animals consists of more than just providing a socializing being. Studies show other benefits of having a pet, such as the positive effect on individuals’ mental and physiological health status. Most research addressing the health benefits of pet ownership show reductions in distress and anxiety, decreases in loneliness and depression, and increases in physical condition [23]. The positive benefits to human health from interacting with animals, focusing on the companion animal, have also be described with the term “zooeyia” [24].

In fact, 63% of owners agreed that having a pet makes them physically healthier, with dog owners more likely to agree, most probably because dog owners exercise more (85% agreeing, compared to 41% of cat owners). Besides, 84% of owners agreed that having a pet makes them mentally healthier. Expressed reasons are the non-judgmental nature of their pets, their playfulness, or physical contact [22]. Another demonstrated positive influence is the blood pressure and heart rate lowering effects that occurs when stroking a friendly-looking dog or even being in the presence of a friendly animal, while it is not necessary to own a pet to obtain these stress-moderating benefits [23]. Many studies have demonstrated the association between pet ownership and cardiovascular health and dog owners appear to have a significantly greater chance of survival after a heart attack compared to people without pets [25,26,27]. Pets can therefore play an important role in reducing absenteeism and visits to family doctors or the hospital [28]. It has been estimated that pet ownership saved Australia $1 billion in 1994 [29], while it may reduce the use of the National Health Service (NHS) in the UK to the value of £2.45 billion per year [30].

Dogs also play an increasing role as co-therapist or as supporter for people with psychological or physical disabilities. The benefits of these animal-assisted activities are improved mood, decreased physiological distress, depression, dementia, and loneliness [31,32]. Examples include resident or visiting dogs in prisons, nursing homes [33], mental institutions, and hospitals where they can reduce patient anxiety in a hospital emergency department [34], reduce pain perceptions in children after surgery [35], or calm young patients at a pediatric dental clinic [36]. Since dogs have extremely sensitive noses, they are used for several purposes such as tracking, bomb detection, and search and rescue. In recent years, canine olfaction has also been more recognized as a diagnostic tool for identifying pre-clinical disease status, such as diabetes (ketones), different forms of cancer, and infections from biological media samples [37].

Animal-assisted therapies can act as co-therapies to facilitate psychotherapy or to provide specific types of therapeutic interventions such as improving motor skills or behavior [23]. Such interventions were effective in improving the state of children or adults with or at risk of developing mental disorders such as attention deficit hyperactivity disorder (ADHD), post-traumatic stress disorder (PTSD), or autism spectrum disorder (ASD) [38,39,40], and for the treatment of PTSD in military veterans [41,42].

Assistance or service animals are trained to perform tasks for the benefit of individuals who have disabilities such as hearing loss, physical disabilities, emotional disabilities, seizure disorders, or diabetes [23].

Finally, a wide range of emotional health benefits from childhood pet ownership has been identified, particularly for those suffering from low self-esteem and loneliness. There is evidence of an association between pet ownership and educational and cognitive benefits, increased social competence, social networks, social interaction, and social play behavior [43,44]. Significantly less absenteeism from school through sickness among children who live with pets has also been reported [13]. Having a dog or cat in the house during the first year of life may protect against childhood asthma and allergy [45,46].

It can therefore be concluded that companion animals contribute significantly toward the public health, but also increasingly, the health of individually challenged persons through animal-assisted interventions [47].

## 5. Negative Aspects of the Human–Animal Relationship

### 5.1. Negative Aspects of Humans for Companion Animals

#### 5.1.1. Changed Feeding Practices

Providing companion animals with feed by humans, has been considered an advantage for companion animals in their relationship with humans. However, the feeding practices can also have a negative impact on companion animals [48,49]. Obesity in cats and dogs is a disease which is rapidly increasing with significant and lifelong implications for animal welfare. Although no universally accepted definition of canine and feline obesity exists, the American Veterinary Medical Association defined obesity being more than 30% above the ideal weight of an animal. Overweight is defined as 10%–20% above the ideal weight. Using Body Condition Scores, it has been estimated that in the United States, 54% of dogs and 59% of cats are obese or overweight. A study in the United Kingdom reported 65% of adult dogs and 37% of juvenile dogs as being obese or overweight [50,51]. Overweight dogs are more likely to be diagnosed with, e.g., urinary tract diseases. Obese and overweight dogs are at risk developing orthopedic disorders and hypothyroidism [52,53]. Obese cats are at higher risk for developing urinary tract disease, diabetes mellitus, and neoplasia [54,55].

Other diet-related problems in companion animals can be caused by the changed feeding behavior of humans, e.g., by providing companion animals with bone and raw feed (BARF) or vegan diets. Risks for companion animals associated with BARF or vegan diets are the presence of microbial hazards, insufficient nutrition, and in raw meat diets the presence of risk materials like thyroid tissue. Through contact with their animals, it is possible that risks could even develop for owners. There could be an increased risk of human salmonellosis because of the presence of *Salmonella* spp. in the diet which can spread to humans through diet leftovers or by contact with animal feces. Recently a review was published on the risks of BARF feeding [56]. The authors concluded that the data for the nutritional, medical, and public health risks of raw feeding are fragmentary, but they are increasingly forming a compelling body of formal scientific evidence. Publications were found reporting the presence of *Escherichia coli O157*, *Salmonella typhimurium*, *Campylobacter* spp., and antibiotic resistant bacteria in the feed. Nutritional problems, such as calcium/phosphorous imbalances and specific vitamin deficiencies [57] are also reported. Moreover, homemade diets are inherently susceptible to nutritional imbalances and deficiencies [58].

Awareness about climate change, public health and animal welfare has incited a major change in dietary choices among many individuals. The number of vegans in the world keeps growing, even quadrupling from 150,000 to 600,000 individuals between 2014 and 2019 in affluent countries such as the UK [59]. The popularity of veganism goes beyond the scope of the human diet, as more people are interested in the possibility of feeding their companion animal a vegan diet than ever before. To create animal-free complete cat food requires replacing nutrients in animal-based materials with plant-based materials. Different sources are used such as corn, rice, peas, soy, potato, and different oils and seeds. Any further nutrients that are missing from plant-based materials, such as taurine and carnitine, are replaced with synthetically produced versions [60,61]. Feeding trials using vegan animal food are either not performed due to testing costs or kept private due to the highly competitive vegan pet food market [62]. Additionally, they reported testing 24 vegetarian diets for cats and dogs and found that one was lacking protein and six did not meet all amino acid concentration requirements. Vegan animal food may not contain meat, but it does contain grains, soy, and corn. Plant-based products, such as grains, can be a source of health problems because of the presence of mycotoxins, for example [63]. Warm, humid storage conditions can lead to the formation of mycotoxins such as aflatoxins, produced by the fungi *Aspergillus flavus* and *Aspergillus parasiticus*. Many regular animal feeds also contain plant-based products, therefore the negative impact of feeding vegan diets to companion animals, especially obligate carnivores such as the cat or the ferret, seems therefore more related to diet insufficiency than to microbial health risks.

Additionally, addressing behavioral problems, the “free” provision of food might fulfil the consumptive part of feeding behavior of our companion animals but does not fulfil the appetitive phase. Especially this phase of feeding patterns can have consequences for the companion animals’ mental health and may result in behavioral problems if appetitive physical and mental challenges remain chronically absent in the human–animal relationship.

#### 5.1.2. Behavioral Problems

In the human–companion animal bond, pets may develop abnormal behavior, including excessive aggression, fear and anxiety, or even abnormal repetitive behavior. Abnormal repetitive behavior (ARBs) were first noticed in zoo-, shelter-, and laboratory animals: All animals housed under stimulus-poor conditions and with limited space. However, companion animals can develop ARBs as well, if the individual’s adaptive capacity is exceeded due to, e.g., a lack of social contact, physical exercise, mental challenges, and in uncontrollable and unpredictable environments (e.g., separation, mistreatment, or inadequate application of cages).

ARBs can either be classified in stereotypies or compulsive disorders [64] with stereotypies generally defined as unvarying repetitive behavior patterns with no obvious goal or function [65]. The terminology of compulsive disorders is preferably chosen for repetitive behavior patterns that are goal-directed and show variability in the repetitive (motor) patterns [66,67]. Under chronic conditions without possibilities to adapt (cope), companion animals may develop stereotypies or compulsive disorders like, e.g., tail chasing, polyphagia, compulsively self-directed licking and/or biting the coat [68], or feather pecking in parrots [69]. Self-directed patterns can result in serious degrees of alopecia, lick granuloma, or even self-inflicted injuries (auto-mutilation) with a risk of infection.

Two main reasons underlie the development of ARBs in our companion animals. First of all, a lot of companion animals are social species eager for social contact. In the human–companion animal bond, the need for social contact with either conspecifics and/or humans [70,71] can remain unfulfilled if owners work from nine to five, five days a week with the pet staying alone at home on a daily basis. On the other hand, a cat which is originally a solitary hunter with a complex dynamic social structure may start overgrooming or house soiling in the presence of another cat in the territory. Such situations might occur in multi-cat households, in the presence of neighboring cats, and in in-stable grouped housing conditions in shelters [72,73].

Secondly, most companion animals are species that are eager for mental and physical challenges on a daily basis. The lack of foraging opportunities, the appetitive phase of feeding behavior [74] might be another reason for the possible development of ARBs in companion animals. Foraging is often regarded as a high priority behavior [75,76], i.e., an internally motivated behavioral pattern that should be performed, or otherwise may induce a state of chronic stress, which may result in behavioral pathology like ARBs as described in many other animal species [77,78,79]. Foraging patterns may include walking, running, jumping, nose pushing, digging, and overseeing the area, all active patterns that imply the daily need for physical exercise and mental challenges in most of our companion animals. Nonetheless, our pets mostly, if not always, get their food for free with minimal foraging challenges, except for going out 3–5 times a day. For some individuals (and especially some dog breeds, e.g., Malinois, border collies, and pit bull terriers) [80,81], situations and contexts with limited challenges can make them more vulnerable to the development of ARBs.

As well as ARBs, other problematic behavior may develop in our companion animals, and the prevalence of some of this behavior is even higher than that of ARBs, for example excessive interspecific and/or intraspecific aggression, fear, and anxiety. At what moment, and which type of problem behavior may develop, depends on the intermingled factors of, e.g., genetics, early life experiences (maternal–child bonding, weaning, socialization [82]), daily environment, and multiple factors in and around the human–companion animal bond.

#### 5.1.3. Breeding and Animal Welfare Problems

The history of breeding animals goes back to a time when humans and animals shared each other’s habitat. Dogs originally have been selectively bred to support human needs, such as hunting, herding, obedience, guarding, rescuing, and for companionship. This artificial selection has generated a large number of dog breeds, displaying a large variation of behavior, size, head shape, coat color, and coat texture [83,84].

Unfortunately, in the last 100 years, intensive selection for extreme looks and a narrow gene pool of many breeds has interfered in the genetic make-up of dogs, leading to unfavorable anatomy (extreme large, or extreme “teacup” small), and genetic predisposition to numerous health, welfare, and behavioral problems [85]. Over 700 inherited disorders and traits have already been described in the domestic dog [86].

One type of dog with a distinct dysmorphology is the brachycephalic dog. Brachycephalic dogs are characterized by a large head and round face due to a shortened muzzle, a high and protruding forehead, and widely spaced large eyes. These facial features fit the concept of baby schema (“Kindchenschema”) proposed by Konrad Lorenz [87]. Infantile (cute) faces are biologically relevant stimuli for rapidly and unconsciously capturing attention and eliciting positive or affectionate behavior, including the willingness to care [88]. The appeal of brachycephalic animals has led to specimens that are the so-called “over-typed” dogs and cats with a too short nose, excessively protruding eyes, too straight angulations, etc. Breeding animals with this type of severe skull and muzzle abnormalities leads to physical and physiological hardship and limits their natural behavior [89,90]. This violates their integrity and is a big risk for their welfare.

Selectively breeding animals in order to express specific traits does not only alter existing animals, but also creates new ones, turning animals into an instrument for human use [91].

The bambino sphynx cat is an example of so called “mutant breeding”, where breeders deliberately stack in two steps the recessive inheriting mutations, which leads to hairlessness in sphynx cats, on the dominant inheriting (lethal) mutation responsible for the shortened legs of the munchkin cat. The lack of hair in combination with short legs interferes with the normal physiology of the cat with regard to the manner of movement, thermoregulation, and skin health. One may argue that artificial selection in exchange for money, status, or aesthetic reasons violates the animal’s dignity and integrity [92]. Conclusively, artificial selection for excessive traits can have direct consequences for individual health and welfare, may obstruct and prevent a pet from fulfilling its behavioral needs, and conflicts with the current moral way of thinking on animal dignity and integrity.

#### 5.1.4. Anthropomorphism

Pet animals are not only perceived in 80%–90% as family members or partners but are almost treated like humans. In one study, up to 62% of owners agreed with the statement “My dog is more important to me than any human being” [93]. This kind of behavior is the result of the attribution of human cognitive processes and emotional states to animals, such as feelings of happiness, love, or guilt. People believe that animals have awareness, thoughts, and feelings. This behavior is called anthropomorphism, personification, or humanization and can also be applied to plants, gods, or objects. Anthropomorphism appears to be caused by the perceived similarity between humans and animals and the extent to which people have developed an affectionate bond with their dogs and cats [94]. Human empathy provides the basis for the attribution of empathy to other animals, as well as attributions of the communicative ability of other animals [95]. Anthropomorphistic behavior can be harmless, such as talking to pets, which many owners do and one of the reasons for this may be the unique ability of humans to recognize facial expressions. Talking to pets is also found to be linked to social intelligence [94].

However, it can lead to animal welfare problems when the feelings of owners no longer match the needs and the intrinsic values of their animal. Examples of this are designer dog clothes, animal perfumes, and jewelry, thought the use of protective coats in colder climates for small, short-haired breeds, e.g., Chihuahuas, is considered useful. The large number of obese pets can also be partly attributed to anthropomorphism. Studies from North America, Europe, and Australia to determine what proportion of animals, mainly dogs, are overweight or obese reported prevalences of between 22%–44% [96]. In 2018, an estimated 60% of cats and 56% of dogs in the USA were overweight or obese [50]. When the owner takes a treat with coffee, it is believed that the dog should also get it. Even chocolate treats are given, when these are potentially fatal for dogs and cats. This also applies to a good meal that is shared with the pet. Dog owners who did not consider obesity to be a disease, maybe because the facial features of their pets fit the concept of baby schema [87], were more likely to have obese dogs [97].

#### 5.1.5. Zooanthroponoses

An often-unrecognized risk for pets is reverse zoonotic disease transmission, the so-called zooanthroponosis. A review on this subject reported 56 articles dealing with human to animal disease transmission [98]. Most of the articles dealt with bacterial pathogens but also viral, parasitical, and fungal pathogens were studied in these publications. Animals reported to have been infected or inoculated with human diseases included wildlife, livestock, companion animals, and other animals or animals not explicitly mentioned. The majority of the studies focused on human to wildlife transmission with an emphasis on *Mycobacterium* spp. For companion animals, MRSA-infection was especially reported but *M. tuberculosis*, influenza A, and *Candida albicans* were also discussed. For all groups of animals, *Microsporum* spp. and *Trichophyton* spp. were identified as infectious agents originating from humans [99].

Recent publications report a different kind of zooanthroponosis: The transmission of high-risk, multidrug-resistant pathogens from humans to animals [100]. A major issue mentioned is the transmission of high-risk clones of extended-spectrum beta-lactamase (ESBL) producing bacteria including *Escherichia coli*, *Enterobacter cloacae*, and *Klebsiella pneumonia* [100,101]. The transmission of carbapenem-resistant NDM-5 producing *E. coli* from previously hospitalized humans to dogs has also been suggested [102]. Transmission of hospital acquired antibiotic resistant bacteria from human patients to their pets has been confirmed, such as the VIM-2 producing *Pseudomonas aeruginosa* ST233 strain in Brazil [103]. This increased transmission of high-risk multidrug-resistant pathogens from humans to animals was related to the closer relationships between humans and companion animals.

### 5.2. Negative Aspects of Companion Animals for Humans

Some authors doubt the generalized pet-effect on human mental and physical health because of conflicting results that are prevalent in this area of science and the lack of publication of negative results [13,104,105,106]. The majority of research evidence was also considered inconclusive due to methodological limitations such as reliance on self-reports, small sample sizes that may not be representative of the general population, homogeneous populations, varying research designs, narrow range of outcome variables that were examined, and the use of cross-sectional designs that do not consider long-term health outcomes [107,108,109].

Other studies found for example that pet ownership was associated with a higher incidence of heart attacks and readmissions in heart attack patients instead of a lower incidence [110] or that pet owners had higher diastolic blood pressure than those without pets [111]. Müllersdorf (2010) showed that pet owners had better general health but suffered more from mental problems such as anxiety, insomnia, and depression, than those who did not own pets [112]. Other studies failed to support earlier findings that pet ownership is associated with a reduced use of general practitioner services [113] or psychological or physical benefits on health for community dwelling older people [114].

Negative effects of pet ownership include dog and cat bites or scratches, the spreading of disease (zoonoses), and fall injuries, caused by falling or tripping over dogs and cats [115]. Allergic reactions may be a consequence of animal contact and affect 15%–30% of individuals (often genetically) predisposed [116]. Allergies relating to more uncommon pets such as fish, birds, and amphibians seem to be increasing in prevalence [117]. Other studies prove that pet ownership in early life did not appear to either increase or reduce the risk of asthma or allergic rhinitis symptoms in children aged 6–10 years. Therefore, advice to avoid or to specifically acquire pets for primary prevention of asthma or allergic rhinitis in children should not be given [118].

#### 5.2.1. Pet-Associated Zoonoses and Their Transmission Routes

There are also less-positive effects that pets can have on health. More excessive forms of anthropomorphism became clear in our study for the presence of zoonotic parasites in healthy dogs and cats. Fifty percent of owners allow pets to lick their faces. Sixty percent of the pets visit the bedroom; 45–60% (dogs–cats) are allowed on the bed, and 18–30% (dogs–cats) sleep with their owner in bed. Six percent of pets always sleep in the bedroom. Of the cats, 45% are allowed to jump onto the kitchen sink [119]. This means that in addition to the detected zoonotic parasites (the hazard), there was a significant potential exposure to these pathogens. In addition to parasites, other pathogens such as bacteria, viruses, and fungi can also be transmitted by animals by direct contact through biting, licking, scratching, sneezing or coughing, handling pets or their body fluids or secretions and by indirect contact through contaminated bedding, food, water, or bites from an arthropod vector [120].

Not every individual will develop symptoms after being infected with a zoonosis. This is the result of various factors such as the causative pet species, housing, the degree of contact and contamination, the ability of a micro-organism to cause disease in humans and animals, but especially due to the degree of immunity of the recipient. In order to assess the risk of disease transmission from pets it is important that the nature and frequency of contacts between pets and their owners or other people are evaluated [120]. We traditionally know that young children (age < 5 years), the elderly (age ≥ 65 years), patients with an impaired immunity, and pregnant women that carry a fragile fetus are at more than average risk of becoming ill after an infection. Moreover, they may have more severe disease, have symptoms for a longer duration, or develop more severe complications compared to other patients. Young children (notably those aged 3–5 years) and some people with developmental disabilities often have suboptimal hygiene practices or higher risk contact with animals which further increases risk [121]. In children, hand-to-mouth behavior is part of their natural development and they mouth their fingers and other objects. In a meta-analysis, the average indoor hand-to-mouth frequency ranged from 6.7 to 28.0 contacts/hour and the average outdoor frequency ranged from 2.9 to 14.5 contacts/hour. The lowest value was attributed to the 6- to 11-year-olds and the highest to the 3- to < 6-month-olds [122]. Fifteen percent of dog owners and 8% of cat owners always wash their hands after contact with the animals [119].

In addition, due to improved healthcare in recent decades, the group of immunocompromised patients has increased sharply. This includes, for example, patients with diabetes, post-splenectomy, after placement of implants and patients being treated with chemotherapy or immunosuppressants. The risk groups are also referred to as YOPIs (young, old, pregnant, and immune suppressed).

Patient surveys and epidemiological studies suggest that the occurrence of pet-associated zoonotic disease is low overall [121]. Many of these pathogens are not reportable and presumably underdiagnosed or not recognized by family doctors due to the general, mostly flu-like, symptoms. Therefore, any reported frequency of such infections is likely underestimated.

To get a better picture of zoonotic risks, a risk analysis is required where a risk score can be calculated using exposure, contagiousness of the infection, and its consequences in the human. In addition, the disease burden of an infection for the population can be calculated and expressed in disability adjusted life years (DALYs). This quantifies the health loss based on two components: The life years lost due to premature death and secondly the proportional loss of quality of life as a result of the disease.

There is a trend towards closer physical contact between owners and their pets or their environment which poses an increased risk of transmission of zoonotic pathogens. These trends (a general direction in which something is developing or changing) will be explained.

#### 5.2.2. Pets in the Bedroom and in Bed

Our publication [119] showed that a high percentage of pets were allowed in the bedroom and in bed, with 6% always sleeping in bed with their owner. Is that a problem? Already from a hygienic point of view, it is not advisable to sleep with animals or take them into bed. They do not pay attention to where they are walking outside and do not wipe their feet after arriving home. Dogs like to roll in carcasses and both dogs and cats regularly lick the anus and thereafter the fur. In a pilot study with 28 healthy dogs and 22 healthy cats that slept with their owners, we tested 68% of the dogs (19) and 32% of the cats (7) positive for Enterobacteriaceae on the fur or footpads. Fleas and flea larvae were found on 14% of pets [123].

As a result of our publication, Chomel investigated in 2011 whether transmission of infections by sleeping with pets and licking the face could be found in literature. He reported that also in the USA, France and the UK, a relatively large number of pets slept in bed with their owner (14%–33% of dogs and 45%–60% of cats) [124]. Similar results of sleeping with pets were also reported from Canada (26% of pets slept with children), the Czech Republic (45%), and Qatar (63.3%) [125,126,127].

Chomel found bacterial infections such as *Yersinia pestis* (plague), *Bartonella henselae* (cat scratch disease), Methicillin-Resistant *Staphylococcus aureus*, and sometimes fatal bite wound infections such as *Capnocytophaga canimorsus* and *Pasteurella multocida*. Furthermore, parasite infections such as *Cheyletiella* spp. were reported.

Feline cowpox is a rare viral infection, but it can be transferred to the human after direct contact. Both animals and humans reveal local exanthema on arms and legs or on the face. In most cases the disease is self-limiting, but immunosuppressed patients can develop a lethal systemic disease resembling smallpox [128].

It can be concluded that, although uncommon with healthy pets, the risk of transmission of zoonotic agents by close contact between pets and their owners through bed sharing is real and has even been documented for life threatening infections such as plague [129,130]. Although pets do not transmit arthropod-borne diseases to people (e.g., Lyme borreliosis, ehrlichiosis, anaplasmosis), they do bring zoonotic disease vectors such as ticks and fleas, in close proximity to people, e.g., when they are sleeping with their animals [121]. While fleas are considered a vector of *Bartonella henselae* (the causative agent of cat scratch disease) tickborne diseases are reported as increasing as ticks expand their ranges [131,132]. With an estimated 65,000 cases a year, Lyme borreliosis is responsible for the largest disease burden of any vector-borne disease in the European Union [133].

Another increased risk associated with close contact with fur is when it is contaminated with zoonotic parasite eggs. Especially with *Echinococcus multilocularis* (fox tapeworm) or *E. granulosus* (hydatid worm, or dog tapeworm). These eggs are immediately infective and may cause serious health problems in the human, many years post infection [134,135]. Despite a low prevalence of infectious (embryonated) eggs of *Toxocara* spp. on dog’s fur, the potential zoonotic risk should not be disregarded [136]. The same risk is applicable for the persistence of sporulated *Toxoplasma gondii* oocysts in dogs’ fur [137].

In relation to this, it is noteworthy that many publications report striking increases of ringworm, a common zoonotic fungal skin infection in mainly children caused by *Microsporum* spp., *Trichophyton* spp. or *Arthroderma* spp., where the presence of pets is always mentioned [94]. However, nowhere has it been suggested that close contact with infected pets in bed increases the infection risk [138,139]. Rodents or rabbits are mainly infected with *T. mentagrophytes*, while *M. canis* is primarily found in dogs and cats. Infection occurs by direct or indirect contact with infected hair, scales, or materials. Infected animals may be asymptomatic carriers without clinical signs. Examples are 16% *M. canis* carriage in a study of European cats [140], 27% in suspected Brazilian cats [141], and the isolation of *T. mentagrophytes* dermatophytes from 4% of clinically healthy rabbits and 17% of guinea pigs in Dutch pet shops [142].

#### 5.2.3. Licking the Face

Licking the face of humans by mainly dogs is an expression of their naturally submissive, positive social behavior. The owner is recognized by the dog as the dominant superior in the ranking. In a pack of dogs, submissive dogs lick their dominant counterparts at the corners of the mouth from a typical submissive attitude [143,144]. Owners apparently allow this as a token of affection from their pet. Such behavior is more common in young animals and has been considered as attention-seeking or care-soliciting gestures. It indicates to the owner the strength of the social bond between dogs and people [120]. Licking or nudging of veterans by service dogs may help take their mind off any negative thoughts, emotions, or memories that they might be experiencing [145].

On the internet, many images can be found of mainly dogs, but also cats and even rats licking their owner’s face. Various studies show that around 40%–50% of owners allow this [119,120]. The question is whether this is harmful due to the potential transmission of infections. The review by Chomel (2011) shows in the literature that infections, especially *Pasteurella* spp. and *Capnocytophaga canimorsus*, were reported to have transmitted to humans by dogs, cats, kittens, and rabbits [124]. *Pasteurella multocida* meningitis has been reported in 36 infants where 87% had been exposed directly or indirectly to the oropharyngeal secretions of household dogs or cats through licking or sniffing [146]. Zoonotic transmission via this route is also assumed for various other pathogens such as gastric *Helicobacter* spp. [147], periodontal pathogens [148], and *Bartonella henselae* the etiological agent in cat scratch disease. The *B. henselae* bacteria may cause ocular complications, including Parinaud oculoglandular syndrome, a severe eye infection. The route of infection is unknown, although direct conjunctival inoculation, most likely with infected flea feces, seems to be most plausible [149]. Knowing, however, that *B. henselae* is present in up to 40% of cat saliva [150], it is more plausible that salivary fluid could be rubbed directly into the eye from the skin after been licked by a cat. There are several anecdotal reports of infections in mainly young children that were transmitted by being licked. One recent example is an 8-month-old baby that presented with fever and preseptal cellulitis with purulent discharge. The causative agent was surprisingly *Corynebacterium bovis*, a bacterium that is normally found in bovine mastitis. It became clear that the dog was frequently allowed to lick the baby’s face and was fed on raw meat [151].

#### 5.2.4. Licking Wounds

There is an ineradicable belief among a large part of the public that the licking of human wounds by dogs can disinfect them and that the saliva thereby has healing properties [152]. In addition, it is regularly reported that a dog’s tongue is believed to be sterile. This is of course not the case and the oral flora of a dog comprises hundreds of species (including pathogenic) bacteria, fungi, and viruses [153,154]. Various wound healing saliva components have indeed been demonstrated in human and animal studies [155,156]. The bactericidal effects of male and female dog saliva facilitate the hygienic function of maternal licking of the mammary and anogenital areas by protecting newborns from fatal coliform enteritis caused by *E. coli* and neonatal septicemia caused by *Streptococcus canis*. However, the saliva is only slightly, and non-significantly, bactericidal against wound bacteria such as coagulase positive staphylococci and *Pseudomonas aeruginosa* [157].

*Capnocytophaga canimorsus* and *Pasteurella multocida* are common commensals in the oral cavity of dogs, cats, and other species [158,159]. Transmission has been reported after the licking of mucous membranes or open wounds [160,161,162]. In patients at high risk, severe wound infections, sepsis, disseminated intravascular coagulation, or death can occur. Patients with immunodeficiency, splenectomy, or alcohol dependence are at a particularly increased risk of infection with *C. canimorsus* [159]. Even immunocompetent persons who have been licked by a dog can develop fatal sepsis [163].

#### 5.2.5. Bite and Scratch Accidents

A further negative effect of companion animal ownership is, of course, accidents inflicted on humans by these animals. These accidents can involve tripping over a cat or being dragged by an enthusiastic dog, but in literature, most evidence points towards biting and scratching incidents. Dog biting and cat scratching incidents can cause physical health problems both at the time of infliction but also afterwards by triggering trauma-related secondary infections. Dog biting incident reports are numerous, and numbers vary from country to country. In the UK, 18.7 dog bites per 1000 population per year were reported [164], while a commission in the Netherlands reported in 2008, 150,000 bite accidents per year in a population of 16 million (9 events per 1000 inhabitants) [165]. Children are especially vulnerable to dog bites. The majority of dog bites occurred in children 5 years of age or younger (68.0%) and almost all (89.8%) of the dogs were known to the children [166]. Recently, a systematic review has been published that analyzed more than 26,000 bites from the literature of the past 30 years about the risk of bites relative to specific breeds of dogs, combining bite incidence with bite severity [167]. The analysis by breed revealed that pit bulls were responsible for the highest percentage of reported bites across all studies (22.5%), followed by mixed breed (21.2%), and German shepherds (17.8%). Dog bite incidents can result in medical treatment, hospitalization and even death. In the Netherlands it was calculated that from the 150,000 dog bite victims per year, around 50,000 seek medical attention and 230 are hospitalized [165].

Between 3% and 30% of dog bites become infected and complications become more severe when infection occurs. More than 100 species of bacteria have been isolated from bacterial infections of dog bites, suggesting that most oral flora of dogs have the potential to be pathogenic [168]. The top 3 pathogens found are *Pasteurella*, *Staphylococcus*, and *Streptococcus*. In literature, specific attention is given to wound infections caused by *Capnocytophaga canimorsus*, because this bacterium is seen as the relatively deadliest pathogen. It was suggested that only 2% of dog bite wounds contained *Capnocytophaga* spp. [154] while others reported infection percentages of 4.2% [169]. Wound infection with this bacterium can lead to severe complications like septicaemia, meningitis, osteomyelitis, peritonitis, endocarditis, pneumonia, purulent arthritis, and disseminated intravascular coagulation. *C. canimorsus* septicaemia has been associated with 30% mortality. The true number of *C. canimorsus* infections is probably largely underestimated due to the fastidious growth of the organism. However, infected dog bites in predisposed persons should be taken seriously especially after splenectomy [170].

Cat bite incidents occur less frequently. In only 5%–10% of reported bite incidents in Australia, cats are to blame. The long incisor teeth inflict less severe superficial wounds but because of the penetrating effect, joint and tendon infections more easily occur. In a review it is reported that 28%–80% of cat bites become infected mostly by *Pasteurella multocida* [169]. The bacterium *Bartonella henselae* can also be transmitted by cats through biting incidents but the transmission of this bacterium is much more related to a cat scratch accident.

Even less frequently reported animal biting incidents are those inflicted by rodents (2% of cases in Australia). These bites have an infection rate of approximately 10%. This could result in rat bite fever in humans, an infection with *Streptobacillus monoliformis* or *Spirillum minus*, characterized by the triad of fever, rash, and arthritis [169].

Cat scratch disease (CSD) was first described in a French boy in 1931 and is a common, often self-limited, disease that usually presents as tender lymphadenopathy caused by *Bartonella henselae* [171]. The cat is considered the primary reservoir for this bacterium, with infected fleas and ticks serving as vectors and humans and dogs as accidental hosts. Vector transmission of this bacterium occurs via two primary routes: Inoculation of *Bartonella* contaminated arthropod feces via animal scratches, most often cat scratches, or by self-inflicted contamination of wounds induced by the host scratching arthropod bites [172]. Immunocompromised human hosts (kidney transplant patients or patients with HIV) are especially susceptible to infection. In these individuals, the disease may be present as a more disseminated form with hepatosplenomegaly, meningoencephalitis, or angiomatosis [173].

#### 5.2.6. Keeping Unusual Exotic Animals

An increasing number of pocket pets and exotic pets are kept by humans. Numbers of ornamental birds in Europe are estimated as 50 million, fish tanks 15.5 million (300 million ornamental fish), small mammals 27 million, and reptiles 8 million [15]. These animal species can also be a source of many zoonotic diseases, especially in young children and immunocompromised individuals. Most cases of these conditions are not serious, and deaths are very rare but some of these diseases can be life threatening, such as rabies, rat bite fever infections, and plague [174].

However, there is also a trend to keep more unusual exotic animals, legally or illegally. These are “wild” animals that are kept in the home such as bats, foxes, skunks, raccoons, meerkats, prairie dogs, kinkajous, sloths, monkeys, apes, prosimians (mammals), parrots, mynah birds, finches (birds), crocodiles, turtles, tortoises, lizards, snakes (reptiles), frogs, toads, newts, salamanders (amphibians), fish, eels, rays (fish), crabs, crayfish, snails, insects, spiders, and millipedes (invertebrates) [175]. Even fruit bats are kept as pets [176] and it is known that bats harbor a higher proportion of zoonoses than all other mammalian orders, including rabies-like viruses that are highly pathogenic for people. [177]. Another example of a zoonosis is monkeypox following the importation of prairie dogs in the USA [178].

Most reptiles and amphibians such as turtles, lizards, and frogs carry *Salmonella* bacteria in their gut, in most cases without visible signs of infection. The infection may cause symptoms of sickness, diarrhea and fever in humans [179]. Zoonotic transmission of *Salmonella* infections causes an estimated 11% of salmonellosis annually in the United States. In cases involving pet turtles, almost half (45%) of infections occurred in children younger than 5 years [180]. *Salmonella* infections are often transferred by feeder rodents [181] and outbreaks highlight the importance of improving public awareness and education in countries who receive imported reptiles [182]. It is advised to exclude reptiles, amphibians, rodents, exotic species, baby poultry, and raw animal-based pet food items from the households of patients at high risk. In lower risk households, an understanding of the risk of salmonellosis and other pet-associated zoonoses and preventive hygiene measures is needed.

The pet trade in general, with its high turnover and diversity of available species, creates a reservoir for pathogens originating from all over the globe. Reptiles account for approximately 10% of live animal shipments imported to the United States [182]. Importation of live reptiles and amphibians for commercial purposes is for a great part unregulated at EU level except CITES and customs regulations [176]. In a risk assessment study, the five pathogens with the highest public health risk caused by the import of exotic animals were *Salmonella* spp., Crimean-Congo hemorrhagic fever virus, West Nile virus, *Yersinia pestis*, and arenaviruses. The risk via legally imported animals was considered low, but substantial for illegally imported animals due to the unknown health status of the animals [183].

Avoiding exposure to exotic pet pathogens in the home is difficult and best achieved by not keeping them in the first place. Otherwise, it is advised to always wash the hands immediately and thoroughly after contact with exotic pets and after handling raw (including frozen or defrosted) mice, rats and chicks. Children should be supervised so that they do not put their mouths close to or kiss exotic animals. Reptiles and other animals should be kept out of rooms in which food is prepared and eaten.

#### 5.2.7. Import of Rescue Dogs and Travelling with Dogs

The number of abandoned and homeless dogs and cats in Europe is estimated to be over 100 million. Countries with more than 1 million abandoned animals are Italy, Romania, Russia, and Ukraine [184]. There is an increasing trend to rescue and import dogs from countries with stray animal problems, in Europe often from Southern or Eastern Europe and in the US from Puerto Rico, the Dominican Republic, Mexico, the Middle East, Turkey, China, and Korea [185]. Many charities and independent groups are involved to rescue dogs and seek adoption in more animal-friendly countries [186]. New owners primarily choose to adopt from abroad based on a desire for a particular dog they had seen advertised and on concern for its situation, while some were motivated by previously having been refused dogs from local rescues [187].

In the USA 85% of all household dogs were neutered and today there are no longer enough dogs being born in the USA annually to replace the approximately 8 million dogs that die each year. Developing countries have hundreds of millions of street dogs available for export, for example Egypt has an estimated 15 million, India, 30 million, and Afghanistan, 100 million [188]. In 2006, more than a decade ago, the Centers for Disease Control (CDC) estimated annual dog imports at around 287,000. Today the number of imported dogs is estimated to be more than a million a year [189].

Imported dogs are reintroducing diseases and parasites that were previously eliminated in the USA [190]. In the USA new and lethal strains of distemper and canine influenza as a result of imported rescue dogs were reported as well as canine brucellosis, rabies, and vector-borne diseases like ehrlichiosis, heartworm, babesiosis, and leishmaniasis [188,191].

In many Southern European countries, there is also a risk of exposure to diseases not encountered in the northern, importing, countries. Animals could be infected with *Anaplasma phagocytophilum, Babesia canis, Brucella canis, Borrelia burgdorferi, Dirofilaria immitis* (heartworm), *Dirofilaria repens* (subcutaneous worm), *Echinococcus multilocularis* (fox tapeworm), *Echinococcus granulosus* (hydatid or dog tapeworm), *Ehrlichia canis, Hepatozoon canis, Leishmania infantum, Linguatula serrata* (tongue worm), *Onchocerca lupi*, rabies, *Rickettsia conorii, Strongyloides stercoralis*, and *Thelazia callipeda*. Except *B. canis, E. canis,* and *H. canis*, the infections are zoonotic and regularly reported [192,193,194,195,196,197,198,199,200,201,202].

Most of these are transmitted by ticks, sand flies, or mosquitoes that are non-endemic in the receiving countries, but a reservoir of infections has been created and the risk is that vectors will become present as result of climate change. [203].

Animals that are infected with rabies or *Echinococcus* spp. may infect people directly. The importation of dogs from endemic, predominantly Mediterranean, regions to Northern Europe, as well as travelling with dogs to these regions carries a significant risk of acquiring an infection. Pet owners are therefore advised not to travel with dogs and to seek the advice of their veterinarian prior to importing a dog from an endemic area or travelling to such areas [193]. For this reason, ESCCAP developed maps on their website featuring European countries and regions with advice on endemic parasites, diseases and recommended treatments when travelling with dogs [204]. A three-year European Union funded project entitled CALLISTO (Companion Animal multisectoriaL interprofessionaL and interdisciplinary Strategic Think tank On zoonoses), has investigated zoonotic infectious diseases transmitted between companion animals and humans and food producing animals [176]. The committee advised that special attention should be given to stray cats and dogs. Stray dogs, in particular, may pose serious health and welfare problems for humans and animals [81], including the transmission of zoonotic diseases such as rabies. Consideration should be given to controlling companion animal movement between areas of the EU endemic for particular zoonoses and areas that are not currently endemic for that disease. Reliable figures are not available, probably because of the fact that many voluntary organizations are responsible for saving and transporting these rescue animals. Such data are essential in order to be able to quantify the actual risks of zoonotic diseases attributable to companion animals and to develop sustainable interventions to prevent transmission to humans and livestock [176].

As long as there are no official guidelines, to prevent the spread of (zoonotic) diseases to new countries, the time, expense, disease risk, and the implications of adopting a dog from abroad should all be carefully considered before importation. As an alternative, the conditions for native dogs could be improved by supporting local charities to organize neutering campaigns and rehoming programs, to build local animal shelters and to improve attitudes towards dogs and their living conditions [205].

#### 5.2.8. Soil Contact

Cats and dogs harbor the enteric nematodes *Toxocara canis* and *Toxocara cati*, and cats are the final host for the protozoal parasite *Toxoplasma gondii*. These parasites can be transmitted to humans because they have an oral–fecal transmission cycle. Humans can be infected by ingestion of infective *Toxocara* spp. eggs or *Toxoplasma oocysts* from contaminated soil (gardens, sandpits, and playgrounds) [206,207]. A recent meta-analysis of data from published records indicates that public places are often heavily contaminated with a pooled global prevalence of *Toxocara* eggs of 21% [208].

Both parasites are considered by CDC as part of five neglected parasitic infections. These infections are considered neglected because relatively little attention has been devoted to their surveillance, prevention, and/or treatment. The diseases that they cause have been targeted as priorities for public health action based on the number of people infected, the severity of the illnesses and the ability to prevent and treat them [209].

Tens of millions of people worldwide are estimated to be exposed to, or infected with, *Toxocara* spp. and recent findings suggest that the effect of toxocarosis on human health is increasing in some countries. Almost one fifth (19%; 1.4 billion individuals) of the world’s human population is seropositive to *Toxocara*. The highest seroprevalence rates were found in Africa (mean: 37.7%) and the lowest in the Eastern Mediterranean region (mean: 8.2%) [210]. *Toxocara* larvae migrate into the body of the human to several organ systems with a preference for the central nervous system (brain, eyes). Human toxocarosis can manifest itself as syndromes known as visceral larva migrans, ocular larva migrans, neurotoxocarosis, and covert or common toxocarosis [211]. Asthma is one of the most common chronic respiratory diseases worldwide, with a negative impact on the quality of life and socio-economic status of patients. Two decades ago, the first evidence was published that suggested that *Toxocara* infection is a neglected risk factor for childhood asthma [212]. The finding that children infected with *Toxocara* spp. are more likely to have asthma compared to non-infected children was recently confirmed in a systematic review and meta-analysis [213]. Cognitive or developmental delays in children or young adults who become infected is of particular concern. Toxocarosis appears to be associated with decreased cognitive function [214,215].

The annual *Toxoplasma* oocyst burden measured in community surveys has been reported as up to 434 oocysts per square foot (4670 per square meter) and is greater in areas with loose soil, that cats like to use to cover their feces in gardens, children’s play areas, and especially sandboxes, also called sandpits and sand piles [207]. Because a single oocyst can possibly cause infection, this oocyst burden represents a major potential public health problem. An estimated one third of the world’s population harbor anti-*Toxoplasma* antibodies. Due to keeping pigs indoors, more education and awareness, the prevalence of the disease in the USA and Europe declined by 50% over the last decades [216]. During an acute invasion of *Toxoplasma* parasites there is mild to major tissue damage without clinical symptoms (latent toxoplasmosis). The most important form is congenital toxoplasmosis when a woman receives her first exposure to *Toxoplasma* during pregnancy. In early pregnancy, this can lead to abortion or to malformations that are not compatible with life shortly after birth. Congenital infections may also be characterized by mental retardation and ocular defects. Acquired infection after birth may result in clinical symptoms such as lymphadenitis, fever, and malaise and probably leads to a clinically symptomatic disease state more frequently than the congenital condition, with an estimated incidence of 30% of all ocular toxoplasmosis cases [216].

Playing in a sandbox is also found to be a predominant risk factor for *S. Typhimurium* salmonellosis in children aged 4–12 years [217]. This can be the result of fecal contamination of the sand by dogs and cats that have been fed raw meat (see Section 5.1.1.).

Young children are especially at risk as they put their hands or other objects in their mouths every 2–3 min [208]. It has also been reported that children ingest a median of 40 mg of soil per day and that one child consumed 5–8 g of soil per day on average [218]. It is therefore advisable not to let children play in public places or on playgrounds with loose sand, but only in sandboxes that can be covered. Furthermore, washing hands after playing outside is important and fingernails must be trimmed to prevent sand being left behind. In this context, a strange trend can be observed as young children play in mud baths on 29 June “as a way to connect and celebrate the natural joys of playing in the mud”. This International Mud Day originated in 2008 and was initiated by an Australian pedagogue who had observed this phenomenon during a visit to Nepal [219]. If the origin of the soil needed for producing the mud is unknown, there is of course an infection risk for the above discussed parasites.

#### 5.2.9. Prevention of the Transmission of Zoonotic Diseases from Pets

To prevent the transmission of zoonotic diseases from pets, risk analysis is of great value. This starts with an assessment of the potential zoonoses in an area, depending on the endemicity (the hazard, H). Hazard characterization also includes prevalence in animals (the reservoir), virulence for man, transmission routes, and survival of the agent in the environment.

The second step is exposure assessment (E). Who is exposed to the potential hazard and for how long or how often? How much of the potential pathogen is needed to become a health risk? This inevitably is directly related to human behavior in relation to their pets.

The third step is to assess the impact of getting infected (I). How serious is the disease, what is the chance of complications, and what economic consequences may be expected (e.g., labor hours lost)? Each of the parameters can be ranked in classes from negligible to the most serious possibility. Ranking is based on literature data, own observations (measuring), or experts’ opinions. The final risk assessment can be achieved by multiplying the outcome of hazard characterization, exposure assessment and impact (H × E × I = a number). The outcome can be compared with other zoonotic agents and a ranking of significance can be made [220].

There is one important parameter to reduce the risk of contracting a zoonotic infection that can directly be influenced, which is exposure. Recommendations are particularly targeted to households with very young children, the elderly, pregnant women, or immunocompromised patients. They are based on reducing exposure to hazards and involve four categories of advice (Table 1).

## 6. Discussion

Reflecting on the changed human–companion animal bond, it can be concluded that pets undoubtedly have a positive effect on human health. Conversely, the human–pet bond seen nowadays is facing many challenges, putting pet welfare under more pressure due to issues such as anthropomorphism, which mainly results in obesity, breeding on extreme appearance rather than health, behavioral problems connected to unfulfilled species specific mental and physical needs, and the provision of inadequate food because owners mistakenly think they feed more naturally.

With regard to the negative effects of pets this article attempts to give an impression of increasing trends in the human–companion relationship that can be observed in society, which appears to increase the risk of transmission of infection between pets and humans. It is mainly a consequence of the increasing contacts between humans and pets and with pathogens secreted by animals in the shared environment. More than 6 out of every 10 known infectious diseases in people can be spread from animals, and 3 out of every 4 new or emerging infectious diseases in people come from animals [222]. The recent pandemic of the Covid19 coronavirus (SARS-CoV-2), that may be originated from bats, is a good example of a recent emerging zoonotic infectious disease. A few cats and dogs have tested positive but are not considered as a source of infection for people.

The proportion of zoonotic human disease that is attributable to pets is largely unknown. Reports about the frequency of such infections are likely underestimated [120]; however, the risk of infection is relatively small for many zoonoses and the severity of the disease is often limited. A person’s age and health status may affect their immune system, and thereby increase his or her chances of getting a disease from animals. Pregnant women should avoid contact with pet rodents, reptiles, cat feces, and raw meat to prevent infection of the unborn child, abortion, or birth defects. If symptoms occur in immunocompetent, non-pregnant persons between 5 and 64 years of age, these are mainly of a general nature such as diarrhea or flu-like symptoms.

Physicians do not regularly ask about the presence of pets or pet contact, nor do they discuss the risks of zoonotic diseases with patients, regardless of the patient’s immune status, which means that many cases of zoonotic diseases go undiagnosed. The general public and people at high risk of pet-associated disease are not aware of the risks associated with high risk pet practices or recommendations to reduce them. Since unfamiliarity with hazards reinforces fear, communication plays an important role in this.

Veterinarians play a key role in education regarding risk reduction by giving advice about responsible pet ownership and the required preventive hygiene. Healthcare providers such as family doctors, school doctors, and pediatricians can also provide information about safe pet ownership. Physicians should ask as part of the medical history about eventual contact with pets, particularly with patients at high risk [120]. The “One Health” initiative aims to reduce this professional gap between vets and physicians [2].

When giving recommendations to prevent zoonotic transmission, one of the often-made remarks is that people nowadays are already too hygienic. It is assumed that there is a protective influence of postnatal infection and that it might be lost in the presence of modern hygiene. This belief is based on the hygiene hypothesis that was formulated in 1989 [223]. It was observed that hay fever in young adults was inversely related to the number of siblings in their family. This hypothesis focused exclusively on allergic conditions as a result of the modern way of life and the assumption that modern hygiene was reducing contact with bacteria. Another view is that some chronic inflammatory disorders have increased over the last decades as a result of decreased frequency of infections due to pathogenic organisms [224]. Another related theory is the “Old friends” mechanism that is based on the positive influence of gut parasites, non-pathogenic environmental bacteria (saprophytes, pseudocommensals), and gut commensals or microbiota. However, decreased exposure to these microorganisms is not the only reason for the increasing frequency of allergies and chronic inflammatory disorders in industrialized countries. Nowadays there is more information about the immunological roles of skin, oral mucosa, and gut microbiota as well as helminths and the influence these have on the immune system [225]. Gut flora may be modified due to diet, obesity, hygiene, antibiotics, but also to psychological stress, vitamin D deficiency, and pollution [225,226,227]. Pollution also has a significant effect on the development of several respiratory problems and diseases. Not only due to outdoor pollution such as fine dust, harmful solids, liquids, or gases [228], but also due to indoor molds as result of insufficient ventilation in energy efficient homes [229]. Finally, there is a clear increase in the allergen production of house dust mites and pollen leading to more exposure and sensitization in susceptible individuals [230]. All together it must be clear that there is much more known about other causes of increasing allergies worldwide than simply excessive cleanliness as suggested in the hygiene hypothesis.

Regarding field infections with helminths such as *Trichuris trichiura* in early life, these are associated with a reduced prevalence of allergies later in life and infants of helminth-infected mothers have a reduced prevalence of eczema. Hookworm infections in developing countries are associated with a reduced prevalence of asthma [231]. The rate of eczema in such countries was found to be about five times higher in infants whose mothers had never had helminths compared with persistent helminth-infected mothers [232,233]. Helminths are nowadays even used under controlled conditions to stimulate immunity. Examples are *Trichuris suis* therapy for Crohn’s disease and *Necator americanus* larvae to treat Crohn’s disease and other autoimmune disorders [234].

There are no clinically apparent childhood infections found to be associated with protection from allergic disorders [235]. It can even result in an opposite effect in the case of *Toxocara* infection by children after soil contact. A recent meta-analysis showed that children infected with *Toxocara* spp. are more likely to have asthma compared to non-infected children [213]. This parasite and asthma both have elevated immunoglobulin E (IgE) levels and eosinophilia in common. That means that precautions should be taken in children to prevent soil contact not only to prevent *Toxocara* infection, but also to prevent acquired ocular toxoplasmosis.

There is no need, therefore, to stimulate the contraction of pathogenic bacteria or helminths to achieve a healthy gut microbiota and to reduce allergic conditions. Recommendations based on the hygiene hypothesis should preferably be based on results from controlled studies to prevent unintentional negative effects.

It is both humans and companion animals who experience negative effects of a changed human–companion animal bond. The education given by vets to their clients should therefore also focus on preventing these negative health and behavioral effects. For instance, by giving science-based advice on feeding practices. In general, regulating authorities should encourage the development of enforcement criteria for breeding dogs and cats to reduce health and welfare risks.

## 7. Conclusions

Pets undoubtedly have a positive effect on human health and well-being, while owners are increasingly aware of pet health, welfare, and well-being. Anthropomorphism, also resulting in behavioral problems and breeding on appearance rather than health, and trends such as keeping exotic animals and importing rescue dogs may result in an increased risk of contracting zoonotic infections.

Recommendations regarding responsible pet ownership, including normal hygienic practices, responsible breeding, feeding, housing, and mental and physical challenges conform the biology of the animal, are key in preventing such negative aspects of the human–animal bond. There is no need to stimulate unhygienic practices following the hygiene hypothesis.

Education can be performed by vets and physicians as part of the One Health concept.

## Figures and Tables

**Table 1 ijerph-17-03789-t001:** Recommendations to prevent the transmission of zoonotic pathogens from pets [121,221].

**Personal Hygiene**
● Washing the hands thoroughly
- after animal contact, at least before eating and drinking and before preparing food or drinks
- after handling raw pet food
- after handling pet habitats or equipment
- after cleaning up feces
- after removing soiled clothes or shoes
● Do not let pets lick the face or open wounds
● Do not feed raw meat to pets
● No pets allowed in the bedroom or in bed
● Keep the nails of children trimmed
● Avoid contact with exotic pets, stray animals, and sick animals
**Keep Pets Healthy**
● Veterinarians should perform an annual health check
● Contact the veterinarian if the animal is thought to be sick
● Keep up to date with the pet’s vaccinations, deworming, and flea and tick control
● Let the health status of imported pets be checked by the veterinarian
**Environmental Hygiene**
● Cover sand boxes so cats do not use them as a litter box
● Wear gloves when gardening or handling soil or sandpits
● Pregnant women: Have someone else empty the litter tray on a daily basis
● Remove pet feces from the yard and public places
**Kitchen Hygiene**
● Cook all meat thoroughly
● Wash all fruit and vegetables thoroughly
● Keep pets and their supplies out of the kitchen
● Do not allow cats on the kitchen sink

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
