# Peer review of "A One Health Perspective on the Human–Companion Animal Relationship with Emphasis on Zoonotic Aspects"

_ijerph, 2020, doi:10.3390/ijerph17113789_

Round 1
Reviewer 1 Report
The authors have taken care to address all relevant comments. I appreciate the attention given to the revision.
Reviewer 2 Report
Hello,
This is a second review of the paper.
This paper must be edited (grammar use, sentence structure, etc.) before published.
Revising the title to show that the focus is on zoonotic aspects is acceptable. Although the focus of the paper remains on the human animal bond in the title. So I am unsure how you focus on the zoonotic aspects of the human animal bond. Maybe human animal cohabitation, relationship, etc. A bond does not have a zoonotic aspect.
Abstract -
"In One Health communications the role” à in complete sentences.
Method: Literature review? The systematic method of reviewing the literature needs to be added in a sentence to reveal the scientific validity of the approach taken.
The sentence starting with 'The importance of companion animals..." it reads as though the focus of the paper is on the negative aspects of the HAB. And it is because of this that that there are behavior problems and a disturbed human-animal relationship and then suseptibility to infections. This reads as the focus of the paper, and therefore should be positioned as such in the abstract. At present, it is not.
Methods: And what numbers did you come up with; how many articles? What years did you review the articles from and to? Noting that the articles are part of a course does not tell us about the methods to acquire the articles.
Again, I understand the inclusion of the positive aspects of the human-animal bond, but the article is not positioned to focus on these. It is positioned, as I read it in the title and subtract and intro paragraphs, to add cautions about the implications of the HAB. So there is a growing HAB, and with that, we need to be aware of zoonosis. I am not denying that that is important, but extensive attention to it in this paper detracts from the point you are trying to make; the main argument.
5. Negative aspects of the HAB. Is it negative aspects of implications of the HAB?
Again, if this section is going to look at the negative aspects of humans for companion animals, this needs to be introduced at the start of the paper as a focus. I strongly encourage you consider what the focus of your paper is, and then keep the literature that is specific to that focus. Right now it reads like a review of anything and everything that the author could think of related to companion animals. Which would be great for a course, but needs focus for a published academic paper. Right now it reads more like a report.
Recommendations should be in the discussion, and flow from the literature reviewed.
When you read the first paragraph of the discussion, this is focused. In fact the entire discussion is focused. The paper should focus on this. Any information provided prior to the discussion that is presented should be directly related to the key points that you are making. The first paragraph is good as a background for your main argument, but it does not need so much attention in the main article.
I hope this is helpful.
Author Response
See the attachment.

This manuscript is a resubmission of an earlier submission. The following is a list of the peer review reports and author responses from that submission.
Round 1
Reviewer 1 Report
This is a very thorough, well-organized paper that leaves no aspect of interaction between humans and companion animals unexamined. It does, however, take an uncritical stance toward the research claiming benefits of pet ownership. In section 4.2 (Negative Aspects), you write, "Some authors doubt the generalised pet-effect on human mental and physical health because of conflicting results that are prevalent in this area of science and the lack of publication of negative results [12,101-103]. The majority of research evidence was also considered inconclusive due to methodological limitations [104]." These two sentences have significant implications for how the preceding claims in the paper should be understood. I'm quite familiar with the literature you cite (Herzog; Wells; et al.), and the doubts about the "pet effect" should receive more attention in your paper. This is especially strking when you report positive effects, but do not mention the methodological problems, such as reliance on self-reports, small sample sizes, homogeneous populations, and varying research designs that often compromise the methodological rigor necessary to understand the therapeutic role of animals.
Along with the tendency to publish only positive results, design problems often lead to conflicting results. Methodological flaws prevent researchers from determining that animals, rather than any other factor, made the difference. For example, consistent, objective measures of the reduction of psychological distress are evasive (see Chur-Hansen et al. 2010).
Moreover, you do not cite studies such as the one that found that pet owners were more likely than non-pet owners were to die within a year of a heart attack (22 percent compared to 14 percent; Parker et al. 2010). Or another that found that pet owners suffered more from psychological problems such as anxiety, insomnia, and depression, than those who did not own pets (Mullersdorf, Granstrom, Sahlqvist, and Tillgren 2010). And I can't locate the reference right now, but a significant number of emergency room visits by the elderly are attributed to falling after tripping over a pet. In sum, would like to see the qualifications to the research of the "pet effect" given more attention. Alternatively, the benefits could be discussed more critically.
Two smaller points:
There is a typo on p. 7, in the heading 4.1.5. "Anthropomorphism" is missing the first "h."
On p. 3, lines 104-107, where you explain the greater number of dogs in the United States, relative to Europe, you write, "Pet owners want to be masters and not servants. Dogs are more controllable 106 than cats, and therefore, U.S. owners feel more psychological ownership over them." My understanding of the Kirk article you cite ("Dogs have Masters, Cats have Staff") is that she's drawing a general conclusion and not one that applies solely to Americans. Yet, that is how you seem to apply this sense of wanting power over animals. If that's not what you mean, this section should be rephrased or omitted.
References
Chur-Hansen, Anna; Stern, Cindy; Winefield, Helen. 2010. “Gaps in the Evidence about Companion Animals and Human Health: Some Suggestions for Progress.” International Journal of Evidence-Based Healthcare 8:140-146.
Müllersdorf, Maria, Fredrik Granström, Lotta Sahlqvist, and Per Tillgren. 2010. “Aspects of Health, Physical/leisure Activities, Work and Sociodemographics associated with Pet Ownership in Sweden.” Scandinavian Journal of Public Health 38:53–63.
Parker, Gordon, Aimee Gayed, Catherine Owen, Matthew Hyett, Therese Hilton, and Gabriella Heruc. 2010. “Survival Following an Acute Coronary Syndrome: A Pet Theory put to the Test.” Acta Psychiatrica Scandinavica 121:65-70.
Author Response
General comments
- Line 49: The first Edition of this book was in 1964
- References 46 and 99 (Association for Pet Obesity Prevention 2019) were double. Ref. 99 has been removed.
- References 78 and 194 (Dietz et al) were double. Reference 194 has been removed.
- Line 545: Typing error, Spirillum is the correct name.
- The name toxocariasis has been replaced by the correct internationally approved name: toxocarosis.
- Line 768: after cleaning up after pets after cleaning up feces
- References 203 and 226 (Aghaei et al) were double. Reference 226 has been removed.
Answers to Reviewer 1
- All revisions in the text are made in blue.
- The methodological limitations and information about negative results of studies that looked for benefits of companion animals for humans are included in the text. The suggested publications are also used as references (Line 350-362).
- The section from lines 104-107 has been removed.
- We adapted the Abstract, the Introduction, and the Discussion (removing text and create paragraph 4.2.9) to create more balance in the article.

Reviewer 2 Report
This is a very fare ranging article, and my suggestion is to focus it for publication. First, to add focus and credibility to the article, a methodology section needs to be added that details how the literature review was done. This is essential. A great deal of work has gone into identifying he articles, and this needs to be detailed. Related, the article needs to open with sharing what the focus of the article is – what countries is the literature referring to (industrialized countries?). For example, you state that 80-90% of pets are seen as family members (reference 12) – where? And what companion animals are being referred to (it starts by sharing cats and dogs, and not horses, (line 84) but other animals are referred to throughout, including a section on exotic animals). I would also suggest limiting the article to one focus – zoonosis. As it is, it is very long and the discussion is only specific to this, so why are the benefits of animals to human life discussed as much as it is? Related, if you are going to speak to the benefits, the one health concept of zooeyia must be presented for this (Hodgson and Darling). I would suggest just having a paragraph or two recognizing this (zooeyia), but you do not need to go into details like you do – it detracts from the focus of your paper. The abstract should be revised with more specific attention to the aim of the paper (you make zoonotic specific recommendations). Overall, the paper reads rather unfocussed, rather than comprehensive review – it is simply too broad. Again, your primary aims is on zoonosis risk with companion animals, so stick to this and briefly introduce zooeyia, increasing pet ownership, etc. to set the stage for the paper. And remain focused on the main argument of the paper. A very minor example, Line 96-107 – I would suggest deleting this, it is not important to the paper to know why you think one animal may be more popular. Most important is the info shared in the paragraph above, because it starts to share the population you are interested in. Another example, Line 119-124 – again, this paragraph seems so far reaching and overgeneralizing. It’s important to know what countries you are speaking to. Also, be careful in the use of language (Line 160), and specifically ‘using’ the animals. There is lots of information in this paper, probably two papers could be written – one focusing on risks and one on benefits.
Author Response
General comments to reviewers
- Line 49: The first Edition of this book was in 1964
- References 46 and 99 (Association for Pet Obesity Prevention 2019) were double. Ref. 99 has been removed.
- References 78 and 194 (Dietz et al) were double. Reference 194 has been removed.
- Line 545: Typing error, Spirillum is the correct name.
- The name toxocariasis has been replaced by the correct internationally approved name: toxocarosis.
- Line 768: after cleaning up after pets after cleaning up feces
- References 203 and 226 (Aghaei et al) were double. Reference 226 has been removed.
Answers to Reviewer 2
- All revisions in the text are made in blue.
- A methodology section has been added in the Introduction.
- The focus of the article is emphasized in the Abstract and the Introduction
- The countries and animals that are involved are better specified in the Introduction
- The one health concept of zooeyia and the publication of Hodgson and Darling has been mentioned in the text (Line 137-138).
- Remarks about animals that may be more popular are removed from the text.
- The countries that are involved are specified (Line 120).
- The use of language, and specifically ‘using’ the animals has been amended (line 162).
- We adapted the Abstract, the Introduction, and the Discussion (removing text from the Discussion and use it as paragraph 4.2.9) to create more balance in the article.
- Regarding the suggestion that two papers could be written – one focusing on risks and one on benefits:
We totally agree that the main focus of this article is zoonotic aspects. This is also the case for the post-graduate course on which this paper is based.
However, discussing only the positive aspects of human contact on companion animals along with pet-associated zoonoses as negative aspects would result in an unbalanced paper.
Therefore, we have well-advisedly chosen to present a complete overview including the most relevant positive and negative aspects for humans as well as for animals.
Splitting this manuscript into two papers would, in our opinion, result in losing the overview of these close-related aspects.
